# LASER: Script Execution by Autonomous Agents for On-demand Traffic Simulation

## Abstract

Autonomous Driving Systems (ADS) require diverse and safety-critical traffic scenarios for effective training and testing, but the existing data generation methods struggle to provide flexibility and scalability. We propose LASER, a novel framework that leverage large language models (LLMs) to conduct traffic simulations based on natural language inputs. The framework operates in two stages: it first generates scripts from user-provided descriptions and then executes them using autonomous agents in real time. Validated in the CARLA simulator, LASER successfully generates complex, on-demand driving scenarios, significantly improving ADS training and testing data generation.

*To make a great film, you need three things–the script, the script and the script.*

*– Alfred Hitchcock*

## 1 Introduction

With the deep learning breakthrough, Autonomous Driving Systems (ADS) have made significant advancements in tasks such as occupancy prediction (Huang et al., 2023b; Wei et al., 2023b), trajectory prediction (Vishnu et al., 2023; Park et al., 2024; Gu et al., 2024), semantic scene completion (Li et al., 2023; Jiang et al., 2024; Cao et al., 2024) and world model (Wang et al., 2023b; Zhao et al., 2024; Wang et al., 2024; Wen et al., 2024). For instance, Tesla's Full Self-Driving (FSD) demonstrates exceptional performance in both common and complex tasks such as lane change, turn, merge, fork and detour for most types of roads including even curvy highways and roundabouts (Tesla, 2024).

The rapid development of end-to-end ADS research and application heavily relies on vast high-quality driving data. On one hand, a large dataset containing multi-modal data from all kinds of sensors (such as cameras, lidar and radar) is essential to train the underlying deep neural network (DNN) models. Taking FSD as an example, Tesla claims their camera-only models are trained with over 160 billion frames of driving data sampled from real-world scenarios, synthetic scenarios generated by simulators, as well as that from other sources (Tesla, 2023). On the other hand, ADS are highly safety-critical, which are required to be thoroughly tested under diverse scenarios, especially some rare, unanticipated ones, to guarantee their capability of handling emergence and avoiding accidents (Wikipedia, 2018).

The most straightforward approach to building datasets for ADS training/testing is to collect real-world traffic data through sensors such as vehicle cameras, which naturally reflect real distributions of the data and can be scaled up through crowdsourcing (Caesar et al., 2020; 2021). However, this method is inefficient, as daily traffic often yields repetitive, trivial scenarios, while safety-critical events, which are rare and high-risk, are seldom included in the training set and thus hardly learned by the model (Feng et al., 2021). Additionally, static data, such as vehicles captured in each frame, lack the flexibility to interact with or manipulate, preventing effective training/testing specific or customized scenarios. Last but not least, online, interactive testing of ADS requires the actors (e.g., vehicles, pedestrians) to be reactive to the behavior of others, which is virtually infeasible for those collected from traffic data.

To address these limitations, another class of approaches is to generate the scenarios from traffic simulators, collecting synthetic data through high-fidelity sensors (Caesar et al., 2020; 2021; Wei

et al., 2024; Zhang et al., 2023; 2024; Suo et al., 2021; Salzmann et al., 2020). These methods allow for creating customized driving scenarios tailored to specific needs, resulting in a controllable and editable dataset for training and testing models. Furthermore, they enable the rapid execution of thousands of diverse and targeted online tests by deploying virtual vehicles attached with trained models in the simulator, facilitating the identification of pitfalls and the resolution of exceptions before costly real-world settings.

Undoubtedly, generating driving scenarios with real-world traffic flows is demanding. Each dynamic object–regardless of a vehicle, bicycle, or pedestrian–exhibits its own time-varying motion patterns, which are often interdependent with those of other objects. Mainstream methods for traffic simulation can be categorized as *rule-based* or *learning-based*. Rule-based traffic simulation employs analytical models to control vehicle movements (Lopez et al., 2018; Casas et al., 2010; Fellendorf & Vortisch, 2010), typically relying on fixed, predefined routes. This approach often results in highly repetitive scenarios with limited behavioral diversity.

In contrast, learning-based methods aim to replicate human trajectories from real-world driving logs to produce varied and realistic behaviors, which leverage techniques–such as imitation learning (IL), reinforcement learning (RL), deep learning (DL) and deep generative models–to generate diverse and realistic driving behaviors by utilizing real-world driving logs as demonstrations. We refer the readers to (Chen et al., 2024) for a comprehensive survey. However, these methods generally face significant challenges in accurately modeling and generating human driving behaviors, often resulting in simplistic actions such as passing or merging (Suo et al., 2021). This is primarily due to three reasons. (a) **Limited and biased training data.** Existing methods commonly rely on datasets such as Nuscenes, which contains only 1000 videos (Caesar et al., 2020). Learning high-level driving behaviors in complex, multi-agent environments must tackle the combinatorial explosion of input states, but the limited, imbalanced nature of the data makes hard to generalize to rare or unseen scenarios, commonly referred to as "long-tail" cases (Chen et al., 2024). (b) **Lack of alignment with human understanding.** The behaviors generated by these models are often not aligned with natural language descriptions or human common sense, making them less interpretable and harder to customize for specific driving behaviors. (c) **Scenario generation in real-time.** When generating interactive traffic scenarios online, these methods typically operate in an auto-regressive manner, where each step's prediction builds upon the previous one. Without goal-oriented guidance, this approach can lead to trivial or ineffective behaviors, and the accumulation of prediction errors may result in catastrophic failures such as collisions or vehicles driving off-road (Xu et al., 2023).

Very recently, deep generative model-based methods (Wang et al., 2023b; Zhao et al., 2024; Wang et al., 2024; Wen et al., 2024) provide a promising way for generating customized traffic data with world models. However, it remains to be a difficult task to generate *interactive* traffic scenarios with diverse, on-demand behaviors.

Recently, large language models (LLMs) and multi-modal language models (MMLMs) have demonstrated remarkable capabilities in common-sense reasoning, planning, interaction and decision-making, showcasing great potential to address the challenges mentioned above (Dubey et al., 2024; Huang et al., 2023a). We propose a new traffic simulation framework, named LASER (**L**LM-based scen**A**rio **S**cript g**E**nerator and Executo**R**) that leverages LLMs to create both intricate and interactive driving scenarios by generating readable scripts to guide step-by-step execution of each dynamic objects within the scenarios, which only requires simple natural language descriptions from the users in the first place.

As illustrated in Figure 1, we first translate user requirements to a master script, which then is converted to sub-scripts for each dynamic object, i.e. each executing actor in the scenario. Based on the rich domain-specific knowledge and advanced reasoning capability of LLM4AD (Wen et al., 2023; Shao et al., 2024), each actor's lifespan is managed by an LLM-controlled autonomous agent that executes its sub-script in real-time. These agents make decisions about intermediate actions based on the current state of the environment at each simulation timestamp, all aiming to achieve their individual goals within the expected time frame.

Leveraging the common-sense and behavior understanding of LLM-controlled agents, we can perform top-down behavior-to-action script interpretation, as opposed to the previous bottom-up action-to-behavior accumulation. The LLM interpretation aligns language-specified behaviors with low-level actions, provides interpretability for the generation process and enables the on-demand gener-

ation of specific long-tail scenarios. With scripts highlighting their agendas, these LLM-controlled autonomous agents cooperate to achieve the tasks, generating on-demand behavior while avoiding accumulated prediction errors.

We design a task set consisting of 17 user requirements encompassing both long-tail and reasonable safety-critical scenarios. We evaluate LASER for on-demand script generation and execution on these tasks in the CARLA simulator (Dosovitskiy et al., 2017). The experimental results demonstrate that LASER can generate scripts based on user requirements effectively, with only 3.18% of the characters in the resulting executable script being inputted by the user to fulfill their demands. The experimental results also show that our approach can execute the script effectively and efficiently, with an average success rate of 90.48%, and usage of 1606.09 tokens per simulation second per agent. Furthermore, manual inspection confirms that our approach can successfully simulate various safety-critical scenarios which can be applied to ADS testing.

In summary, the primary contribution of our work is to propose an on-demand, interactive approach for traffic simulation, which includes a script generator and LLM-controlled autonomous agents as the executor. To the best of our knowledge, this is the first time to achieve on-demand scenario generation in ADS testing.

## 2 RELATED WORKS

We have covered the related work on learning-based traffic simulation in Section 1. A recent trend in autonomous driving is to leverage LLMs which have demonstrated exceptional capabilities in human-like tasks such as common-sense understanding, planning, decision-making, and interaction (Dubey et al., 2024; Huang et al., 2023a), trained on vast datasets of trillions of tokens and images from the web. These models exhibit a deep reservoir of actionable knowledge, which can be harnessed for robotic manipulation through reasoning and planning (Wen et al., 2023). Recent research has explored the use of LLMs to develop autonomous agents that execute natural language tasks in interactive environments (Driess et al., 2023; Brohan et al., 2023; Belkhale et al., 2024). A notable example is RT-H, which enhances agent robustness and flexibility by decomposing high-level tasks into sequences of fine-grained behaviors, referred to as "language motions" (e.g., "move arm forward" followed by "grasp the can"). This approach effectively leverages multi-task datasets, significantly improving performance (Belkhale et al., 2024).

Substantial efforts have also been directed towards integrating LLMs with ADS, underscoring the models' superior abilities in understanding and decision-making within driving scenarios (Sharan et al., 2023; Renz et al., 2024; Shao et al., 2024; Wen et al., 2023). For example, CarLLaVA achieved first place in the sensor track of the CARLA Autonomous Driving Challenge 2.0, surpassing the next-best submission by 32.6%. This success is attributed to its integration of the vision encoder LLaVA with the LLaMA architecture as its backbone (Renz et al., 2024). Additionally, LLM-Assist outperformed all existing learning- and rule-based methods across most metrics in the Nuplan dataset by leveraging LLMs' common-sense reasoning to refine plans generated by rule-based planners (Sharan et al., 2023).

Further research has demonstrated the capability of LLM-integrated ADS to execute tasks based on natural language instructions, revealing its potential for modeling complex human driving behaviors (Wang et al., 2023a; Shao et al., 2024). For instance, LMDrive showed that LLM-controlled driving agents could interpret and follow high-level driving commands, such as "Turn right at the next intersection," by aligning these instructions with vehicle control signals using a vision-language model as the foundation (Shao et al., 2024). More recently, DiLu demonstrated that with few-shot learning, LLMs could achieve results comparable to RL-based planners, significantly reducing the computational cost of deploying multiple LLM-controlled agents simultaneously (Wen et al., 2023).

## 3 METHODOLOGY

### 3.1 FRAMEWORK

To achieve on-demand and interactive traffic simulation, we propose a framework called **LASER**, as illustrated in Figure 1. LASER consists of two stages, implemented by two modules respectively,

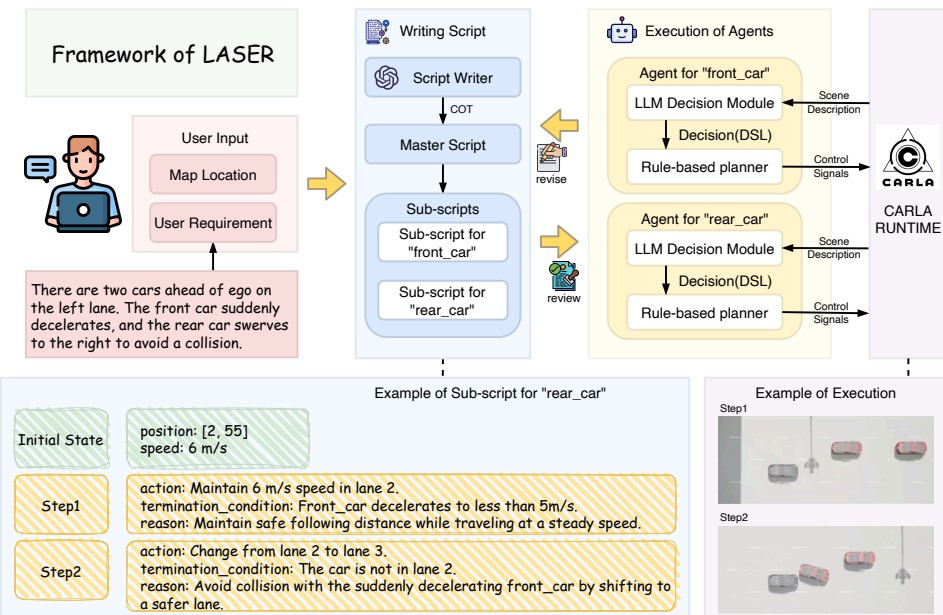

Figure 1: Framework of LASER

i.e., script writer (Section 3.2) and LASER-Agent (Section 3.3). Unlike the previous learning-based methods that conduct generation and simulation simultaneously, our LASER framework first generates scripts that define logic-chained behaviors (LCB) with natural language instructions from the user requirements. It then executes the script by the real-time cooperation of LASER-Agents.

The two-stage framework detaches actor behavior from the scenario with a natural language script. Compared to road data collection that collects on-road scenarios at the state level, our LASER framework records scenarios at the behavioral level, which enables dynamic and flexible execution during the simulation. Compared to learning-based traffic simulation that generates scenarios at the state level, LASER generates scenarios at the behavioral level, this enables on-demand generation and top-down execution of language-specified behavior, and easy editing on actor behavior to improve performance.

To have a glance at the complexity of behaviors that LASER can simulate, we present an example of a user-defined scenario in Figure 1. The user requests a safety-critical scenario on the highway: Two cars are driving in the left lane ahead of the ego vehicle, which is in the right lane. Suddenly, the front car decelerates due to a mechanical failure. The rear car, unable to brake in time, switches to the right lane to avoid colliding with the front car, unaware that the ego vehicle is behind it. In this situation, the ego is responsible for quickly recognizing the potential for the rear car to change lanes due to the front car's abrupt deceleration and must react by decelerating promptly to prevent a collision. We will demonstrate detailed implementations of the script writer and LASER-Agent modules with the example shown in Figure 1 in the following sections.

## 3.2 SCRIPT WRITER

Directly executing user requirements with autonomous agents is unstable. User requirements can be ambiguous, leading to varying behaviors across different execution attempts. Additionally, fulfilling the user requirement often requires coordination among multiple autonomous agents. Without a shared consensus on how to achieve this goal, these agents may act independently, potentially resulting in failure.

To address these challenges, we use script writer to translate user requirements into scripts. To ensure consistency across different execution attempts, the behaviors in the script must be detailed and concrete, with sub-scripts that contain detailed LCB instructions for each agent to execute. To

ensure effective coordination, these sub-scripts must align under a single master script, serving as a unified consensus for all agents.

In our work, the generation of scripts follows a hierarchical chain-of-thought (CoT) manner (Wei et al., 2023a) to enhance the common-sense reasoning and planning ability of LLM as GPT-4o does (OpenAI, 2024). The script writer first generates a master script that outlines the framework of the story, then further generates sub-scripts that contain detailed LCB instructions for each actor based on the master script. This ensures behavior consistency over agents and executions. The scripts are written in natural language, to enable easy editing of the behaviors. The generating procedure of the master script and sub-scripts for individual agents are shown in the following paragraphs.

**Master-script Generation.** Based on the user's initial requirements and (optionally) a map description to specify the surrounding layout, the script writer generates a master script using a CoT approach. First, the script writer prompts the LLM to generate a story that aligns with the user's requirements. The LLM then outlines key stages in the sequence of events, starting with the initial state, where each stage acts as a direct cause or prerequisite for the next. For example, given the user requirements illustrated in Figure 1, the following master script is produced:

> 1. Initial state: The Vehicle Under Test (VUT) is in the rightmost lane, with two cars ahead in the leftmost lane.
>
> 2. Stage 1: The front car in the leftmost lane suddenly decelerates. Reasoning: The deceleration may be due to an obstacle or the need to reduce speed significantly for an intersection.
>
> 3. Stage 2: The rear car in the leftmost lane swerves into the rightmost lane. Reasoning: The rear car swerves to avoid a collision with the front car.

**Sub-scripts Generation.** Based on the master script, the script writer queries the LLM to generate sub-scripts containing detailed, step-by-step LCB instructions for each individual actor. These sub-scripts clearly outline the specific actions each actor must perform, using natural language to chain behaviors logically. Each action is paired with a termination condition, specifying when the task is complete and when the next action should begin, along with the reasoning behind it. This ensures that the behavior not only aligns with the overall narrative but also allows for flexible execution by the autonomous agent.

The resulting sub-scripts are structured similarly to movie scripts. They begin with an initial state, defining the actor type (e.g., truck, car, or pedestrian), and then break down into several sequential steps. Each step includes an action, termination conditions and a reason, all of which work together to link the actor's behavior with logical decision-making. An example of a sub-script is shown in Figure 1. The initial state specifies the actor's lateral and longitudinal position, as well as its initial speed. The action defines a simple, concrete motion that can be easily executed by our LASER-Agent, such as merging into the leftmost lane. The termination condition outlines measurable criteria, which may depend on the actor's own behavior or interactions with others, for example, when the longitudinal distance to another vehicle is within 2 meters. Finally, the reason clarifies the rationale behind each action, enhancing the agent's understanding and enabling more flexible execution, especially during interactions with other actors.

## 3.3 LASER-AGENT

The second stage of grounding the sub-scripts into execution is achieved through the collaboration of LASER-Agents. To facilitate the comprehension of language instructions and behavior, we employ LLM-controlled driving agents. These agents, designed to operate autonomously, execute the sub-scripts step by step based on real-time environmental observations, working together to bring the entire scenario to life. Since fine-tuning LLMs with vehicle control signals and applying LLMs to learning-based planners both require substantial computational resources during runtime (especially when managing multiple agents), we integrate each agent with an LLM-based decision module alongside a rule-based planner. The LLM-based decision module, equipped with common-sense, plays a crucial role in converting language-based LCB instructions into executable actions. Every 0.5 seconds, the LASER-Agent encodes the environmental scenario into a descriptive format, integrates

sub-script LCB instructions to create a prompt, and queries the LLM for an executable decision. This decision is then carried out by the rule-based planner.

**LLM-based Decision Module.** This module processes the scenario description along with LCB sub-script instructions, generating an executable decision every 0.5 seconds. It begins by checking if the termination condition for the current step is met. If the step is complete, it transitions to the next one. The module then predicts an executable decision based on the step's instructions and the scenario description, which includes parameters such as target speed, lane change direction and lane change delay.

LLM brings a common-sense understanding to translate language instructions to executable decisions. Instead of directly outputting vehicle control signals where LLMs do not excel, this higher-level decision-making process offers better alignment with the model. This approach enables zero-shot grounding of language-specified behaviors more effectively.

To enhance LLM's comprehension of the current traffic environment, we encode the surrounding traffic conditions into a standardized textual scenario description, following the approach outlined in DiLu (Wen et al., 2023). This description includes all essential information for decision-making, such as the number of available lanes, the positions, speeds and lane-change statuses of both the subject vehicle and surrounding vehicles.

**Rule-based Planner.** It outputs vehicle control signals to execute the decisions made by the LLM-based decision module at every frame. It tracks a path consisting of waypoints on the map and uses PID control (Wikipedia, 2023) to regulate speed. Whenever the rule-based planner receives a new decision from the LLM-based decision module, it generates a new path based on the current position, following the lane change direction and the delay specified in the decision. At each frame, the planner tracks the waypoints on the path, while the PID controller calculates vehicle's steering and throttle.

This lightweight design allows the control of multiple agents simultaneously, enabling their complex behaviors and interactions. The rule-based planner functions as a humble executor of the LLM's decisions, without incorporating safety constraints such as maintaining distance from other vehicles. This design ensures that the agent can exhibit alarmingly realistic behaviors.

## 4 EVALUATION

In this section, we present a comprehensive evaluation of script writer and LASER-Agents for on-demand traffic simulation. All code and the results are available on an anonymous repository. [1]

### 4.1 SETTING

**User requirements**. We design 17 scenario generation tasks, each representing a complex traffic execution requirement that is challenging to capture through traditional road collection or existing simulation methods. (cf. Table 3 in Appendix A.1.1 for details). For example, tasks such as "Accident", "Ambulance", and "reckless Driving" present long-tail scenarios that are rarely encountered. "Swerve" and "Three in Line 1" depict reasonable safety-critical situations where even experienced drivers could make mistakes. To navigate these scenarios effectively, one must be able to anticipate signs of an impending accident.

**LASER Setup**. Our experiment utilizes CARLA 0.9.15 (Dosovitskiy et al., 2017), a widely used open-source simulator for closed-loop ADS testing. Built on the UE4 engine, CARLA offers realistic graphics, a variety of vehicle and pedestrian models, and diverse maps. We assess each task across three road segments from Town04 and Town05, conducting 20 simulations per segment for effectiveness (totaling 60 simulations per task) and 5 simulations per segment for efficiency (totaling 15 simulations per task).

All experiments employ GPT-4o as the LLM, which queries every 0.5 seconds. Appendix A.1.3 shows an example of LLM reasoning process. To evaluate interactions between LASER-Agents and ADS in safety-critical scenarios, we use our LASER-Agents to test the end-to-end ADS InterFuser (Shao et al., 2023), which ranked #1 in the CARLA challenge 2022.

---

[1]https://github.com/CXYyp5SkNg/CXYyp5SkNg.github.io

Table 1: Evaluation of user involvement in script generation. (* indicate safety-critical tasks)

| Task | User involvement percentage ↓ |
|---|---|
| Accident | 5.43% |
| Ambulance | 4.41% |
| Caught in Pincer | 0.47% |
| Failed at Start | 7.36% |
| Newbie Lane Change 2 | 0.35% |
| Cut-in* | 1.50% |
| Swerve* | 3.15% |
| Three in Line 1* | 2.74% |
| average | 3.18% |

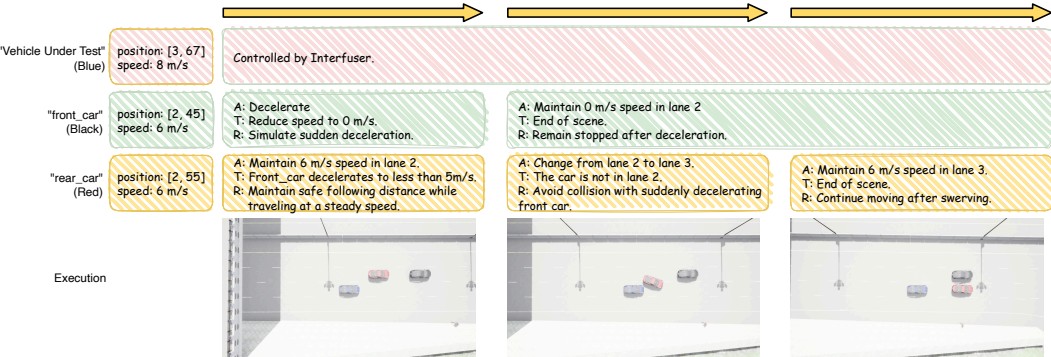

Figure 2: A case of script generation result. (A: Action T: Termination Condition R: Reason)

**Road Segments**. We evaluate each task on three highway segments with varying lane numbers (including one curved segment) and three urban segments (including a curved road) to assess the generalizability of LASER.

**Metrics**. The effectiveness of script writer is measured by the user involvement percentage, defined as the average proportion of user-provided characters (excluding spaces and newlines) in the final executable script. We assess script execution success rate as the number of traffic simulations meeting user requirements divided by the total number of simulations conducted. Generated traffic simulations are manually reviewed against criteria outlined in Table 4 (in Appendix A.1.3). Efficiency is evaluated using token cost, defined as the number of tokens used per simulation second per agent, and time cost, representing the real-world simulation time per simulation second.

## 4.2 EVALUATION ON SCRIPT GENERATION

To evaluate the effectiveness of on-demand script generation, we select eight user requirements related to long-tailed scenarios from the task set. For each task, script writer generates five scripts, which we then manually refine until the scripts successfully fulfill the user requirements.

The experimental results are summarized in Table 1, while Figure 2 visualizes a selected case for the task "Swerve". The results indicate that script writer effectively generates on-demand scripts, with an average user involvement percentage of just 3.18%. Most inaccuracies in script writer 's outputs stem from imprecise numerical values, such as positions and speeds. Additionally, there are instances where script writer overlooks steps implied by the user requirements. For example, in the "Failed at Start" task illustrated in Figure 4 (Appendix Section A.1.2), the bus should initially move in the left lane while the car stops in the right lane, before both vehicles change lanes simultaneously. However, script writer incorrectly bypasses this step, causing the bus to change lanes at the start.

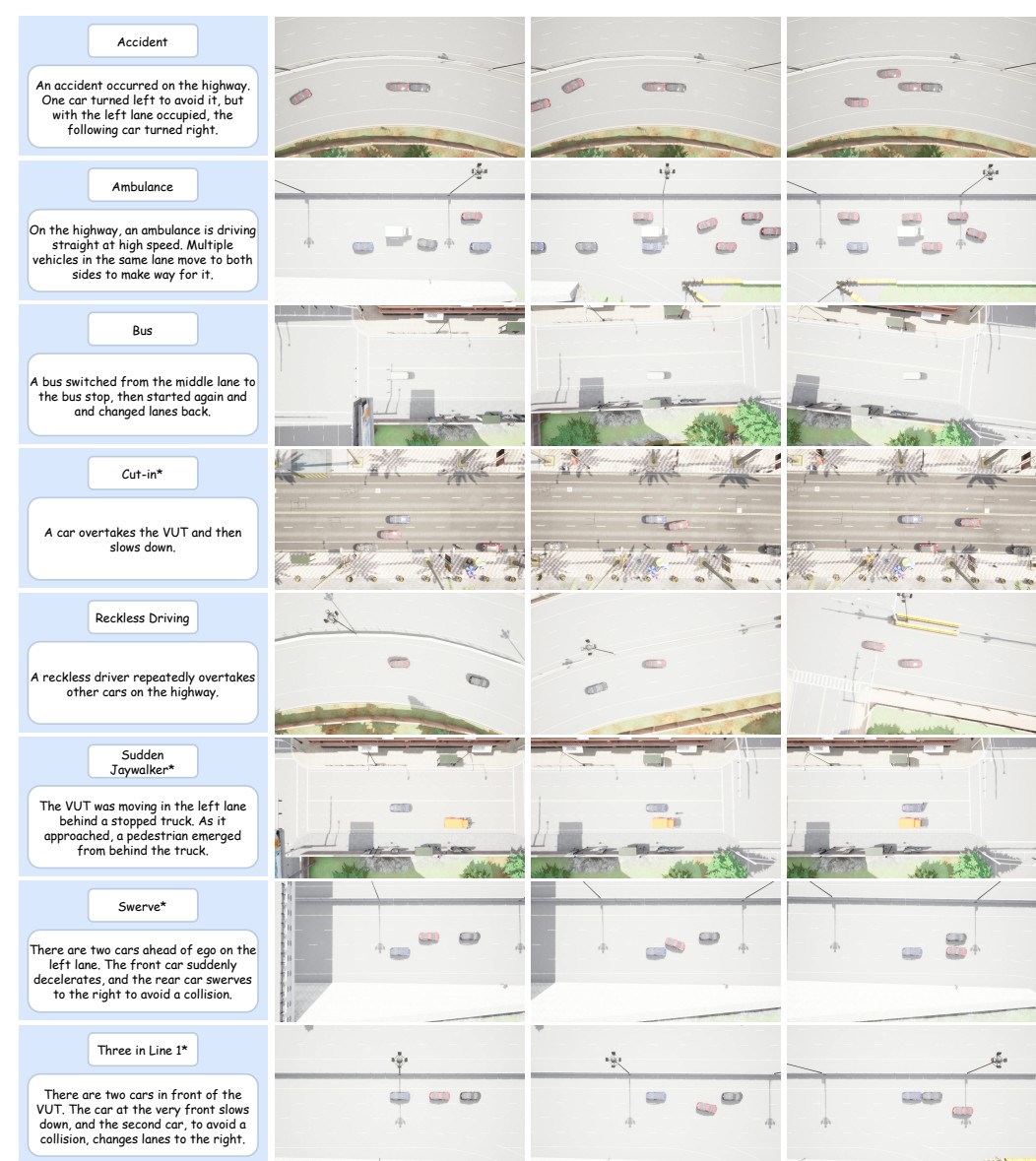

Figure 3: The visualized results for script execution. (* indicate safety-critical tasks)

## 4.3 EVALUATION ON SCRIPT EXECUTION

To evaluate the effectiveness and efficiency of script execution, we select eight user requirements from our task set that involve complex interactions, including four tasks focused on safety-critical scenarios. For each task, script writer generates a script based on the user requirement, which we then manually modify to ensure it meets the requirements fully. Scripts are executed by our LASER-Agents to assess their effectiveness and efficiency.

**Effectiveness.** The experimental results are shown in Table 2. The results indicate that LASER-Agents successfully execute the scripts, achieving an average success rate of 90.48%. Most errors arise from inaccuracies in numerical comparisons, such as positions and speeds, while some mistakes result from hallucination.

**Efficiency.** The efficiency results are presented in Table 2, revealing an average of 1,606.09 tokens and 7.87s for generating a one-second simulation. In the Ambulance task, the inclusion of multiple dummy agents to simulate congestion significantly increased the input tokens needed to describe

Table 2: Results of effectiveness and efficiency for script execution. (* indicates safety-critical tasks)

| Task | Road type | Execution success rate ↑ | Token cost ↓ | Time cost ↓ |
|------|-----------|--------------------------|--------------|-------------|
| Accident | highway | 100% | 1,556.08 | 9.63 |
| Ambulance | highway | 96.43% | 2,387.81 | 15.63 |
| Bus | urban | 91.67% | 1828.40 | 7.02 |
| Reckless Driving | highway | 44.07% | 2333.84 | 7.11 |
| Cut-in* | urban | 98.33% | 1305.83 | 5.76 |
| Sudden Jaywalker* | urban | 100% | 2130.40 | 7.55 |
| Swerve* | highway | 100% | 732.37 | 5.60 |
| Three in Line 1* | highway | 93.33% | 573.99 | 4.62 |
| average | | 90.48% | 1606.09 | 7.87 |

other vehicles' states, leading to a higher token cost. Similarly, the Reckless Driving and Sudden Jaywalker tasks require lengthy scripts for the actors, resulting in elevated input token counts. Using GPT-4o API service, generating a 40-second simulation with 3 LASER-Agents incurs a cost of approximately $1. The majority of the time cost is attributed to querying the LLM, with a 10-second simulation taking around one minute to generate.

## 5    LIMITATIONS

**Manual description of map layout**. LASER's execution relies on manually formatting map layouts for scenario generation. This approach can lead to inaccuracies and inefficiencies, particularly in complex environments. Implementing automated map interpretation (e.g., querying image-to-text models with the initial frame) could greatly enhance the framework's scalability and accuracy.

**Lack of automatic search for scenario details**. The current system necessitates user-in-the-loop revision for scripts. While this allows tailoring the generation that aligns with users' intentions, it restricts the system's capacity to autonomously generate a multitude of test cases with the same initial goals but varying details. Developing a more intelligent script writer capable of automatically searching for reasonable and elaborate scenario details poses a challenge due to the knowledge gap between off-the-shelf LLMs and the specific requirements of the simulation environment.

**Computational overhead for real-time execution**. Integrating LLM-controlled agents with real-time execution in complex environments incurs significant computational overhead, especially when scaling the simulation to multiple agents. Future enhancements could focus on optimizing interactions between the LLM-based decision-making module and the rule-based planner, aiming to reduce latency and computational load while maintaining high performance and decision accuracy.

**Generalization to real-world scenarios**. Although our framework demonstrates strong performance in simulated environments, its ability to generalize to real-world driving scenarios may hinge on the simulation's fidelity. Ensuring that virtual agents accurately mimic human driver behavior across diverse global contexts remains an ongoing challenge.

## 6    CONCLUSION

In this paper, we introduce LASER, a novel approach that leverages LLMs to generate on-demand traffic simulations. Our two-stage framework separates scenario generation from real-time execution, providing greater flexibility, scalability, and customizability compared to traditional simulation methods. By utilizing LLM-controlled agents, LASER offers a more human-like interpretation of driving behaviors, ensuring coherent and realistic interactions within the simulated environment. The experimental results demonstrate that LASER effectively meets diverse user requirements for both general and safety-critical driving scenarios, showcasing high accuracy and adaptability in scenario creation. Overall, the proposed approach represents a significant advancement in on-demand traffic simulation for ADS training and testing.

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

# A APPENDIX

## A.1 EXPERIMENT DETAILS

### A.1.1 TASK SET

We present our task set of 17 user requirements in Table 3.

Table 3: Task set (* indicates safety-critical tasks)

| Task | User requirement |
|---|---|
| 1. Accident | An accident occurred on the highway. One car turned left to avoid it, but with the left lane occupied, the following car turned right. |
| 2. Ambulance | On the highway, an ambulance is driving straight at high speed. Multiple vehicles in the same lane move to both sides to make way for it. |
| 3. Bus | A bus switched from the middle lane to the bus stop, then started again and changed lanes back. |
| 4. Caught in Pincer | A car to the left is overtaking, forcing the ego vehicle to decelerate, while a car behind speeds up, pressuring it to accelerate. |
| 5. Cut-in* | A car overtakes the VUT and then slows down. |
| 6. Failed at Start | A car parked in front of the bus station started, changed lanes, and collided with the bus that was changing lanes to park. |
| 7. Meet in Mind 1 | Two cars simultaneously change lanes to the middle lane. |
| 8. Meet in Mind 2 | Two cars, unable to see each other, simultaneously attempt to overtake a car in the middle. |
| 9. Merge Alternately | Two cars collided in the right lane at a road intersection, and several cars in the rear weaved into the left lane. |
| 10. Newbie Lane Change 1 | A car in front on the right changes lanes without accelerating, causing a collision with the ego vehicle. |
| 11. Newbie Lane Change 2 | A truck is to the left of the ego vehicle, and a car in front on the right changes lanes without accelerating, causing a collision with the ego vehicle. |
| 12. Reckless Driving | A reckless driver repeatedly overtakes other cars on the highway. |
| 13. Sudden Jaywalker* | The VUT was moving in the left lane behind a stopped truck. As it approached, a pedestrian emerged from behind the truck. |
| 14. Surrounded | Four police cars surround the criminal's vehicle from the front, back, left, and right. |
| 15. Swerve* | There are two cars ahead of ego on the left lane. The front car suddenly decelerates, and the rear car swerves to the right to avoid a collision. |
| 16. Three in Line 1* | There are two cars in front of the VUT. The car at the very front slows down, and the second car, to avoid a collision, changes lanes to the right. |
| 17. Three in Line 2 | The front car suddenly hit the stopped car ahead, and the ego car collided with the front car. |

### A.1.2 EXPERIMENT DETAILS OF SCRIPT GENERATION

We present the visualization for the execution of some representative scripts in Figure 4. Further examples are given in Appendix A.2 for readability.

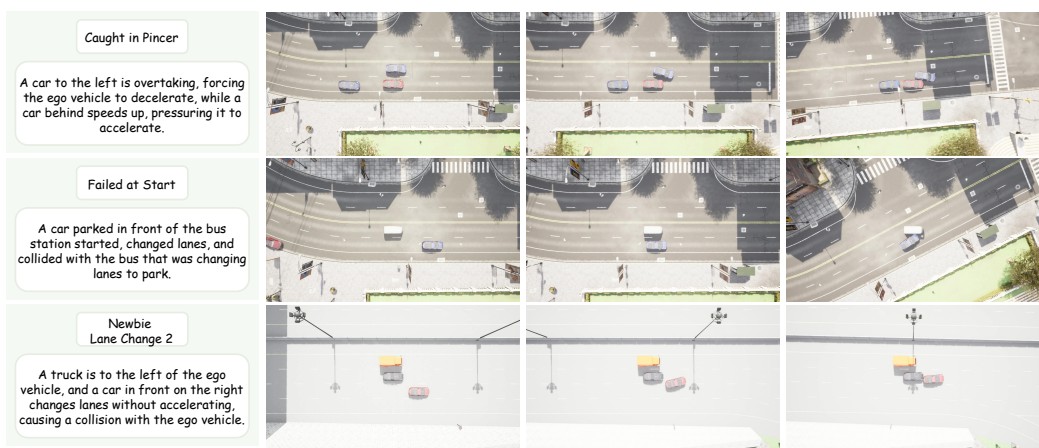

Figure 4: The visualized results for script generation.

### A.1.3 EXPERIMENT DETAILS OF SCRIPT EXECUTION

**Criteria for reviewing simulations.** We present the criteria for reviewing simulations to evaluate the effectiveness of script execution in Table 4.

Table 4: Criteria for reviewing simulations (* indicate safety-critical tasks)

| | |
|---|---|
| Accident | The two cars correctly change lanes according to their scripts before colliding with the stopped accident vehicles. |
| Ambulance | As the ambulance approaches, vehicles in the same lane move aside one by one to make way, without any collisions. |
| Bus | The bus changes lanes and stops at the bus station within a ±2.5m range, then starts again and changes lanes back. |
| Reckless Driving | The reckless car successfully overtakes the first car from the left lane and the second car from the right lane. |
| Cut-in* | The car successfully overtakes the front car then slows down. |
| Sudden Jaywalker* | The jaywalker steps out from behind the truck as the ego vehicle approaches. |
| Swerve* | The front car decelerates to a stop. The rear car changes into the ego vehicle's lane, nearly hitting the ego vehicle before colliding with the front car. |
| Three in Line 1* | The front car decelerates to a stop, while the rear car changes lanes to the right before colliding with the front car, and then drives away. |

**Example of LLM decision module's reasoning.** Take the "rear_car" in the "Swerve" task as an example. Following the script outlined in Figure 2, the agent maintains its speed while cruising on the highway. Once the "front_car" decelerates below 5 m/s, the agent immediately changes lanes to the right. To achieve this, we encode the sub-script and the state of the "front_car" in the user prompt and query the integrated LLM. The LLM automatically evaluates whether the termination condition has been met based on the state of the "front_car" and makes decisions according to the current action specified in the script.

### A.2 ADDITIONAL TASKS

We present visualized results for additional tasks in our task set in Figure 5.

**System prompt**

You are a driving assistant in a simulated scene to help us generate dangerous scenes to test autonomous driving systems.

You must follow the steps given by user to generate dangerous scenes. To do this, you can drive alarmingly and ignore traffic rules.

Every 0.5s, you will be given:

Steps: Steps to be taken to accomplish your task.

Previous step: The step you were taking in the last 0.5s.

Observations: The location, speed, and acceleration of you and other vehicles in the 2D plane. for example: location=[106.0, 3.0] $m$ means the vehicle's longitudinal position on the lane is 106.0 $m$ and its lateral position from the leftmost lane center is 3.0 $m$.

You should response me step by step:

1. Previous Step Evaluation: Assess the completion status of the previous step based on observations and termination condition.

2. Previous Step Status: Completed/Incomplete

3. Your Current Step: Step you think should be taken based on your current observations. Move to next step if you think the last step has been completed. For example: "Current step: step i. ...".

Finally: Execute actions for the current frame by a tool call.

**User prompt before "front_car" decelerates**

Steps:

Step 1. action: Maintain 6 m/s speed in lane 2, termination_condition: Front_car decelerates to less than 5 m/s.

Step 2. action: change from lane 2 to lane 3, termination_condition: The car is not in lane 2.

Step 3. action: Maintain 6 m/s speed in lane 3, termination_condition: Hold for the next 10 seconds

Last step: Step 1. action: Maintain 6 m/s speed in lane 2, termination_condition: Front_car decelerates to less than 5 m/s.

Observation:

You are driving on a road with 4 lanes, and you are currently driving in the second lane from the left. Your current position is `(98.54, 3.50)`, where 98.54 is the longitudinal position and 3.50 is the lateral position. The longitudinal position is parallel to the lane, and the lateral position is perpendicular to the lane. Your current speed is 5.84 m/s, acceleration is 0.73 m/s^2, and lane position is 98.54 m.

- `front_car` is driving in the same lane as you and is ahead of you. Its current position is `(108.83, 3.50)`, where 108.83 is the longitudinal position and 3.50 is the lateral position. The longitudinal position is parallel to the lane, and the lateral position is perpendicular to the lane. Its current speed is 5.81 m/s, acceleration is 1.02 m/s^2, and lane position is 108.83 m.

- `VUT` is driving on the lane to your right and is behind of you. Its current position is `(89.27, 7.00)`, where 89.27 is the longitudinal position and 7.00 is the lateral position. The longitudinal position is parallel to the lane, and the lateral position is perpendicular to the lane. Its current speed is 7.88 m/s, acceleration is 3.73 m/s^2, and lane position is 89.27 m.

**LLM reasoning before "front_car" decelerates**

1. Previous Step Evaluation: We are maintaining a speed close to 6 m/s in lane 2. The front car in the same lane is driving at 5.81 m/s and accelerating.
2. Previous Step Status: Incomplete
3. Your Current Step: Step 1. Maintain 6 m/s speed in lane 2.
Executing actions for the current frame:
- Maintain a target speed of 6 m/s
- Follow the current lane (lane 2)
Let's proceed with these actions.

**LLM decision before "front_car" decelerates**

current_step_number: 1
lane_change_direction: FOLLOW LANE
lane_change_delay: 0
target_speed: 6

**User prompt after "front_car" decelerates**

Steps:
Step 1. action: Maintain 6 m/s speed in lane 2, termination_condition: Front_car decelerates to less than 5 m/s.
Step 2. action: change from lane 2 to lane 3, termination_condition: The car is not in lane 2.
Step 3. action: Maintain 6 m/s speed in lane 3, termination_condition: Hold for the next 10 seconds
Last step: Step 1. action: Maintain 6 m/s speed in lane 2, termination_condition: Front_car decelerates to less than 5 m/s.
Observation:
You are driving on a road with 4 lanes, and you are currently driving in the second lane from the left. Your current position is `(104.42, 3.50)`, where 104.42 is the longitudinal position and 3.50 is the lateral position. The longitudinal position is parallel to the lane, and the lateral position is perpendicular to the lane. Your current speed is 5.76 m/s, acceleration is 0.70 m/s^2, and lane position is 104.42 m.
- `front_car` is driving on the same lane as you and is ahead of you. Its current position is `(114.07, 3.50)`, where 114.07 is the longitudinal position and 3.50 is the lateral position. The longitudinal position is parallel to the lane, and the lateral position is perpendicular to the lane. Its current speed is 4.65 m/s, acceleration is 1.24 m/s^2, and lane position is 114.07 m.
- `VUT` is driving on the lane to your right and is behind of you. Its current position is `(97.10, 7.02)`, where 97.10 is the longitudinal position and 7.02 is the lateral position. The longitudinal position is parallel to the lane, and the lateral position is perpendicular to the lane. Its current speed is 7.99 m/s, acceleration is 0.65 m/s^2, and lane position is 97.10 m.

**LLM reasoning after "front_car" decelerates**

### Previous Step Evaluation:
- **Previous Step**: Maintain 6 m/s speed in lane 2.
- **Observations**:
- Your current speed is 5.76 m/s and front_car's speed is 4.65 m/s.
- front_car's speed is less than 5 m/s, which meets the termination condition.
### Previous Step Status:
- **Status**: Completed
### Your Current Step:
- **Current Step**: Step 2. Change from lane 2 to lane 3.
Executing actions for the current frame:
### Actions:
1. **Current Step Number**: 2
2. **Lane Change Direction**: Right Lane Change
3. **Lane Change Delay**: 0.1 s
4. **Target Speed**: Maintain 6 m/s
Executing the action.

**LLM decision after "front_car" decelerates**

current_step_number: 2
lane_change_direction: RIGHT LANE CHANGE
lane_change_delay: 0.1
target_speed: 6

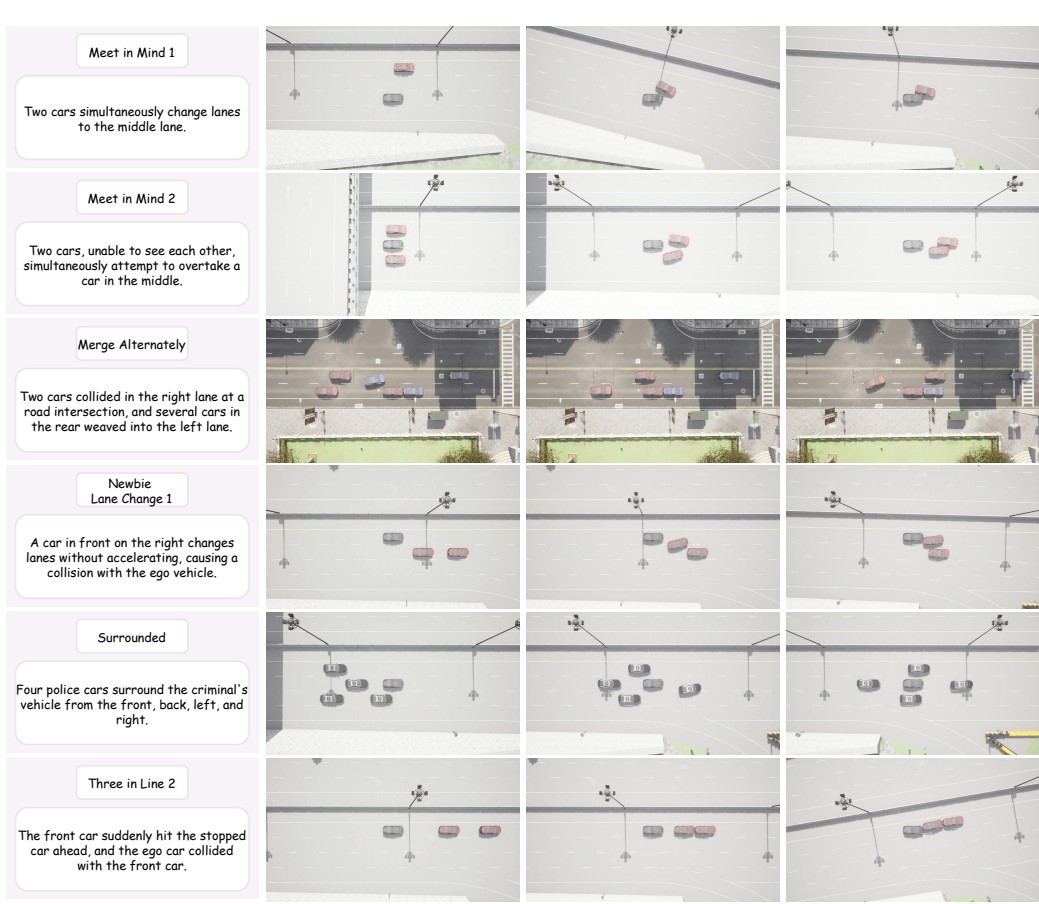

Figure 5: The visualized results for additional tasks.

