# OpenReview forum: "LASER: Script Execution by Autonomous Agents for On-demand Traffic Simulation"
_ICLR.cc/2025/Conference — Submitted to ICLR 2025_

### Official Review · Reviewer_kKJ4 · 2024-10-24

**Soundness:** 3
**Presentation:** 3
**Contribution:** 2
**Rating:** 5
**Confidence:** 4

**Summary:**

This paper proposes a framework named LASER that generates traffic simulation conditioned on language description. First, an LLM is called to create scripts for the simulation, including a high-level script for the whole scene and sub-scripts for individual agents including detailed task descriptions.

Then an autonomous agent executes the scripts by invoking LLM to give high-level commands and following the command with a rule-based controller.

The simulation is carried out in Carla. The metrics used to measure success are 1) the percentage of human modification in the final executable script, 2) the number of tokens and wall clock time for executing a 1-second simulation 3) the success rate of the simulation measured by whether the agents follow the language description.

**Strengths:**

1. I think the application of LLM to translate language descriptions of traffic scenarios into detailed scripts is interesting and reasonable
2. The simulation is carried out in closed-loop, which adds to the fidelity of the result
3. The paper is overall well-written and easy to follow

**Weaknesses:**

1. I think the fact that humans need to keep modifying the scripts generated by LLM is concerning. It's unclear whether the use of LLM actually saves time.
2. Following the previous comment, the metric of Effectiveness seems very objective, i.e., what if one person is better than another person at modifying the LLM-generated scripts? I think a more reasonable metric is to measure how often the script can work without any human modification.
3. There lacks comparison to some existing works such as LCTGen, CTG.

**Questions:**

1. How is the observation generated? Is it heuristic?
2. Is there communication between the autonomous agents during execution? In cases where one agent's action is dependent on other agents, is it all resolved in the script-writing phase, i.e., everything has to be specifically written down as scripts?
3. What happens if one of the agents is a self-driving car following some policy? In the case where all cars are controlled by LASER, I don't see much benefit in closed-loop simulation compared to open-loop generation.

---

### Official Review · Reviewer_z4Vh · 2024-10-25

**Soundness:** 1
**Presentation:** 3
**Contribution:** 1
**Rating:** 3
**Confidence:** 4

**Summary:**

This paper introduces LASER which leverages ChatGPT-4o to generate traffic simulations. The method contains two stages: converting user description into a step-by-step plan for each agent and then using LLM-controlled agents to convert the plan into executable waypoints for a low-level PID controller. To evaluate LASER, the authors evaluate the success execution rates of 8 scenarios.

**Strengths:**

+the application can be potentially useful for augmenting traffic data.

+the paper is overall clear and provides relevant details of the proposed approach.

**Weaknesses:**

-missing related work on using LLM on generating traffic behavior e.g., [1] [2] [3].

-the proposed approach is incremental compared with existing work (e.g. [3]).

-scripts still require manual revision and thus greatly limit its scalability. The number of total generated evaluation scenarios is limited.

-diversity of the generated scenarios is not evaluated.

-no evaluation on the added values of the generated scenarios has been conducted.

-compared with learning based controllers, using a PID controller to directly control a vehicle can differ from human driving behavior (e.g., in terms of acceleration, jerk, etc.)

[1] Language Conditioned Traffic Generation, CoRL 2023
[2] Language-Guided Traffic Simulation via Scene-Level Diffusion, CoRL 2023
[3] Promptable Closed-loop Traffic Simulation, CoRL 2024

**Questions:**

Can you respond to each of the weaknesses point I mentioned?

---

### Official Review · Reviewer_HruR · 2024-11-03

**Soundness:** 2
**Presentation:** 3
**Contribution:** 2
**Rating:** 3
**Confidence:** 5

**Summary:**

This paper identifies the problem that existing scenario generation methods struggle to provide flexibility and scalability, which may limit the effective training and testing of autonomous vehicles. Therefore, the authors propose LASER, a novel framework that leverage large language models to conduct traffic simulations based on natural language inputs. Specifically, there are two stages during the generation: (1) it first generates scripts from user-provided descriptions and then executes them using rule-based planner in real time.

**Strengths:**

1.	The idea of using LLM to aid the generation of scenarios is interesting and promising. Since LLMs have general knowledge about driving and traffic, they should help generate risky and long-tail scenarios. This gives them the potential to overcome problems that exists in pure data-driven scenario generation methods.
2.	The paper is generally well-written and well-organized. Figure 1 clearly shows the components and pipeline of the proposed method.

**Weaknesses:**

1.	The 4th and 5th paragraphs in the introduction section discuss literature of traffic scenario generation in simulation. However, a lot of relevant works are missing. I understand that it is not necessary to review related works in the introduction section, but this review is also missing in the related work section with “We have covered the related work on learning-based traffic simulation in Section”. The authors can find more relevant papers in the following survey papers [3][4][5]. For example, there are recent papers that using LLM to generate scenarios [1][2], which I believe are highly related to this paper.
2.	Based on my understanding, the proposed method is very similar to LCTGen [1], which converts natural language to some intermedium representation (the script in this paper) and then execute the script. The main difference is that this paper uses a rule-based planner while LCTGen uses a learned trajectory generator. In addition, this paper uses Carla as the sensor simulator while LCTGen only considers 2D simulation with BEV representation. Considering the similarity and differences, I feel the novelty of the proposed method is somehow limited.
3.	As a generation task, the quantitative evaluation is important. In this paper, I think only two main metrics are considered, namely User involvement percentage and Execution success rate. However, I think they are mainly correlated to user instructions and prompt design. We still need metrics to evaluate the realism, diversity, and instruction following capability of the method, which I believe are important for indicate whether the generated scenarios are useful or not.
4.	The current system requires human users in the loop to revise the script. I appreciate that the authors mentioned this in the limitation section. This seems a strong limitation since the labor requires from human may reduce the benefit of using this system. A human user may directly use similar API interface and tools to quickly compose a scenario without using LLM.
5.	It is not clear how the generated scenarios benefit the testing of autonomous vehicles. Either using the scenarios to train a policy or evaluate a trained policy would provide insights on the value of the proposed system. Since this paper uses Carla, which provides high-quality sensor simulation, it would be good to see how some end-to-end driving model perform in the generated scenarios.

---
[1] Tan, Shuhan, Boris Ivanovic, Xinshuo Weng, Marco Pavone, and Philipp Kraehenbuehl. "Language conditioned traffic generation." arXiv preprint arXiv:2307.07947 (2023).

[2] Zhong, Ziyuan, Davis Rempe, Yuxiao Chen, Boris Ivanovic, Yulong Cao, Danfei Xu, Marco Pavone, and Baishakhi Ray. "Language-guided traffic simulation via scene-level diffusion." In Conference on Robot Learning, pp. 144-177. PMLR, 2023.

[3] Ding, Wenhao, Chejian Xu, Mansur Arief, Haohong Lin, Bo Li, and Ding Zhao. "A survey on safety-critical driving scenario generation—A methodological perspective." IEEE Transactions on Intelligent Transportation Systems 24, no. 7 (2023): 6971-6988.

[4] Zhong, Ziyuan, Yun Tang, Yuan Zhou, Vania de Oliveira Neves, Yang Liu, and Baishakhi Ray. "A survey on scenario-based testing for automated driving systems in high-fidelity simulation." arXiv preprint arXiv:2112.00964 (2021).

[5] Schütt, Barbara, Joshua Ransiek, Thilo Braun, and Eric Sax. "1001 ways of scenario generation for testing of self-driving cars: A survey." In 2023 IEEE Intelligent Vehicles Symposium (IV), pp. 1-8. IEEE, 2023.

**Questions:**

1.	How do the authors design the task set in table 3? Is there any justification for selecting these tasks about the coverage and risk level?
2.	Which LLM does the authors use? Did the authors try different LLMs and see a large performance gap between them?

---

### Official Review · Reviewer_v7Ws · 2024-11-04

**Soundness:** 1
**Presentation:** 2
**Contribution:** 1
**Rating:** 3
**Confidence:** 4

**Summary:**

This paper introduces LASER, an LLM-powered approach for dynamic traffic simulation, allowing users to create complex, interactive driving scenarios using simple natural language prompts. LASER translates these prompts into master scripts, which are then broken down into sub-scripts for each actor in the scenario. LLM-controlled agents manage these actors, making real-time decisions based on environment states to achieve specific goals. Tested on the CARLA simulator, LASER achieved a 90.48% success rate with minimal user input. The proposed method enables on-demand, safety-critical scenario generation for ADS testing, marking a first in on-demand scenario creation in autonomous driving simulations.

**Strengths:**

- The paper is easy to follow.

**Weaknesses:**

- The proposed method is not fully automated and still requires human in the loop.
- Defining user involvment with average proportion of user-provided characters does not fully make sense since users may need to read through a lot more before even provide sensible and effective modification. In such a case, the actual user involvment may be greatly underestimated.
- Execution success rate in table 2 is defined as the scripts being successfully executed. However, it doesn't indicate whether the generated simulation actually makes sense or is useful.
- There is no baseline at all and it is hard to judge how effective the proposed method is.
- The technical contribution of the paper is extremely limited. Further addressing some of the mentioned limitation (e.g., manual description of map, lack of automatic search) may be a good direction.
- The number of generated cases (17) is quite limited. It is hard to (i) be convincing that the proposed method can be to certain degree scalable and (ii) have the statistics of quantitative results sufficiently convincing.

**Questions:**

- Missing references regarding natural language to driving behaviors: [R1][R2][R3], especially R3 is extremely relevant in the sense of using LLMs to write code for traffic simulation. It would be nice to also include some of them as baselines for comparison.
- The metric of user involvement doesn't seem convincing. Could the authors provide other metrics that better capture user involvment? Perhaps something like engagement time for user to make the modification, number of issues fixed, task completion rate, etc.
- Does the user get to run the simulation when "debugging" the generated script?
- In table 2, could the authors provide more informative metric other than execution success rate? Perhaps something that measures the generated simulation actually being intended (e.g., the simulated scenario actually involve a swerving car leading to the ego-car changing its behavior or even crashing).
- There is no baseline. [R3] can be a suitable baseline for comparison as this paper follows extremely similar setup. Others like [R1][R2] can also be potential baselines as they take language as inputs. Some of the aspects to be compared can be diversity of the scenarios, how safety critical can the simulated environments be, number of simulation, etc.

Reference

[R1] Language conditioned traffic generation

[R2] Language-guided traffic simulation via scene-level diffusion

[R3] Text-to-Drive: Diverse Driving Behavior Synthesis via Large Language Models

---

### Meta-Review · Area_Chair_TSLv · 2024-12-16

**Metareview:**

This paper presents LASER, a framework that leverages large language models (LLMs) to generate traffic simulation scenarios from natural language descriptions. The approach has two stages: first generating scripts from user descriptions, then executing them with autonomous agents in real-time in the CARLA simulator.
Strengths:
+ The application of LLMs to traffic scenario generation is a potentially useful direction
+ The paper is generally clear and well-organized
+ The closed-loop implementation in CARLA adds realism

Major Weaknesses:
- Limited novelty compared to existing work like Text-to-Drive and LCTGen that also use LLMs for traffic scenario generation
- High reliance on manual script revision undermines the claimed benefits of automation
- Very limited evaluation (only 17 scenarios) with no comparison to baselines
- Lack of metrics for scenario quality, diversity and usefulness
- Insufficient analysis of how the generated scenarios benefit autonomous vehicle testing

The paper has significant limitations in technical novelty, methodology and evaluation. Multiple reviewers noted strong similarities to existing work and raised concerns about manual intervention requirements that limit scalability. The evaluation lacks meaningful comparison to relevant baselines and provides limited evidence of practical value. Given these substantial weaknesses and no author response addressing these concerns, I recommend rejecting this paper.

**Additional Comments On Reviewer Discussion:**

The reviewers raised consistent concerns about novelty, evaluation rigor, and practical limitations that were not addressed through author responses or revisions. Without engagement from the authors to address these fundamental issues, the identified weaknesses remain valid reasons for rejection.

---

### Decision · Program_Chairs · 2025-01-22

Reject